# Analysis of ADAS Radars with Electronic Warfare Perspective

**DOI:** 10.3390/s22166142

**Published:** 2022-08-17

**Authors:** Alper Cemil, Mehmet Ünlü

**Affiliations:** 1Electrical and Electronics, Engineering Department, Ankara Yıldırım Beyazıt University, 06760 Ankara, Turkey; 2Electrical and Electronics, Engineering Department, TOBB University of Economics and Technology, 06510 Ankara, Turkey

**Keywords:** ADAS radars, electronic warfare, intentional jamming, coherent jamming, ADAS security, DRFM

## Abstract

The increasing demand in the development of autonomous driving systems makes the employment of automotive radars unavoidable. Such a motivation for the demonstration of fully-autonomous vehicles brings the challenge of secure driving under high traffic jam conditions. In this paper, we present the investigation of Advanced Driver Assistance Systems (ADAS) radars from the perspective of electronic warfare (EW). Close to real life, four ADAS jamming scenarios have been defined. Considering these scenarios, the necessary jamming power to jam ADAS radars is calculated. The required jamming Effective Radiated Power (ERP) is −2 dBm to 40 dBm depending on the jamming scenario. These ERP values are very low and easily realizable. Moreover, the effect of the jamming has been investigated on the radar detection at radar Range Doppler Map (RDM) and 2-Dimensional Constant False Alarm Rate (2D-CFAR). Furthermore, the possible jamming system requirements have been investigated. It is noted that the required jamming system will not require high-end technology. It is concluded that for the security of automotive driving, the ADAS radar manufacturer should consider the intentional jamming and related Electronic Counter Countermeasures (ECCM) features in the design of ADAS radars.

## 1. Introduction

Radar, which is an acronym for “Radio Detection and Ranging”, is a detection system that uses radio waves to estimate the position and velocity of a target, approximately. The application areas of the state-of-the-art radar systems include military applications, civilian aviation, navigation, mapping, meteorology, radio astronomy, and medicine [1].

One of the most important application areas of radar systems in daily life is automotive radars. The automotive radars, together with the other sensors such as light detection and ranging (LIDAR), ultrasound, and cameras, constitute the fundamentals of the Advanced Driver Assistance Systems (ADAS) and self-driving automobiles [2]. The millimeter-wave (mm-wave) radar systems provide better performance in all weather conditions than the other sensors used in ADAS [3,4]. For the sake of secure driving and parking, the automotive radars are the critical element for the detection of objects and obstacles, defining their position and measuring the relative speeds [5]. Radars, which are capable of providing long-distance detection, allow safe driving opportunity under risky conditions that decrease visibility of the environment [4,6]. However, such a motivation for the development of automotive radars also requires the development of low-cost and compact radar systems, which can be installed on the vehicles easily reaching a robust performance under all types of working conditions. For this purpose, the silicon technology has solved the problem of compact and high-performance system implementation of mm-wave radar systems [7,8].

Fast-chirp/Chirp-sequence Frequency Modulated Continuous Wave (FMCW) waveform with Multiple Input Multiple Output (MIMO) techniques are commonly used in the automotive radars [9]. ADAS radar systems are one of the radar systems that use these techniques and generally operate in 24 GHz and 77 GHz to achieve higher velocity and range resolution [5]. The technological advancements in Complementary Metal-Oxide-Semiconductor (CMOS) Integrated Circuits (IC) that may utilize both analog and digital circuitry allow demonstration of low-cost and compact radar-on-chip and antenna-on-chip systems [3].

With the recent developments, the employment of the affordable millimeter-wave sensors that operate at 77 GHz is expected to become more frequent than ever [3]. Nevertheless, this brings the challenge of interference management to avoid any risk in high traffic jam conditions. The interference problem is addressed by the key players in automotive radar’s Original Equipment Manufacturers (OEM). Moreover, there is a growing interest in intentional jamming of ADAS radars in the literature [10,11,12,13,14,15]. Security risks and concerns of sensors used for ADAS were raised, and the possibility of an ADAS radar jamming on a Tesla Model S was experimentally examined [10]. In [11], a simple low-cost spoofing device based on modulated backscatter was proposed to demonstrate the ADAS jamming successfully. In [12] the jammer, based on Software Defined Radio, is used to analyse the spoofing attacks against ADAS radars. The designed jammer successfully spoofed the ADAS radars. In [13], the vulnerability of ADAS sensors to spoofing attacks is addressed. Furthermore, the detection and mitigation method for the spoofing attack is proposed by applying multiple beamforming. In [14], the importance of security for wireless sensing and radio environment is highlighted. The study also includes the jamming of ADAS radars. In [15], the security issues of ADAS radars have been examined considering noise jamming and spoofing. Considering all these studies, although the ADAS radars are examined for different jamming conditions, none of these studies have investigated the ADAS radar for the possible, real-life scenarios for on-purpose jamming. More importantly, the studies do not examine under what conditions the on-purpose jamming can be attained. Hence, a detailed investigation is needed, where the jamming scenarios with possible real-life scenarios are considered in order to determine under what conditions and system parameters the jamming can be achieved.

In our study, compared to other studies in [10,11,12,13,14,15], we analyse the ADAS radar jamming with the traditional Electronic Warfare (EW) perspective. The aim of this study is to take an attention on intentional jamming of ADAS radars. In this paper, we investigated the ADAS jamming topic with close to real life scenarios, calculated the necessary Effective Radiated Power (ERP) depending on scenarios, and further analysed the effects of jamming on radar detection. While defining the scenarios and jamming engagements envelope, the features of the ADAS radars such as beamwidth, spatial resolution, and max range have been considered. For example, because of the spatial resolution capability of LRR, the scenario is defined without angle separation. In other words, the real echo and ghost echo are in the line of sight of the jammer and LRR. Moreover, we present the investigation of performance measures of ADAS radars considering different aspects of jamming systems such as power levels, antenna, and waveform generators with Digital Radio Frequency Memory (DRFM). The detail of the proposed results is given in the following sections of the paper.

## 2. Intentional Jamming of ADAS Sensors

Despite the fact that the effort in IC technologies pushes the limitations to develop low-cost and compact radar systems, the performance of the ADAS systems as a measure of secure driving operation is discussed in the following section of this paper.

### 2.1. Security of ADAS Sensors

ADAS sensors and their functions are described in [2]. The recent developments in the IC technology and motivation to develop fully autonomous vehicles increases the necessity of investigation of possible security issues in automotive radars. An ADAS consists of different automotive radars which are used for different purposes. A possible security concern for the automotive radars could be intentional jamming. Intentional jamming is one of the important parts of EW. The goal of this study is to investigate automotive radars from the EW perspective. We try to find out whether ADAS radars can be jammed and what could be the key performance feature of the required jamming system. If jamming is possible, then the necessity of implementation of expensive Electronic Counter Countermeasures (ECCM) features, like military radars, for the ADAS radars will be analysed.

EW has been used extensively in the Defense Industry where there are military standard and military unique requirements are considered. Furthermore, the amount of equipment is limited, and the cost of each equipment is cheaper than the development cost. So, each equipment should be qualified thoroughly. On the other hand, automotive radars are produced in mass numbers and the cost of each item is an important factor for the automotive industry.

There are several studies on unintentional jamming, namely interference, of the ADAS radars. One of the important references on this topic is the MOSARIM (More Safety for All by Radar Interference Mitigation) project report [16]. In the MOSARIM project, interference scenarios and mitigation measures have been investigated thoroughly and evaluated with respect to their applicability and expected efficiency in automotive radars. Example of studies on intentional jamming can be found in [10,11,12,13,14,15]. In [10], Tesla Model S (Bosch MRR4) was jammed with fix frequency noise, swept frequency noise, spoofing techniques, and they observed the results from the vehicle display. With the noise jamming techniques, the front car cannot be detected. With the spoofing techniques, radar provides different range information. ADAS radar spoofing also investigated in [11,12,13,14,15] and it is observed that the spoofing techniques would be the most effective jamming type.

The intentional jamming might be considered as an extraordinary case. However, as the use of autonomous automobiles becomes more common in our life, it might also be considered by some malicious people/organization to harm someone.

### 2.2. Electronic Warfare

As radars started to be used for military purposes, the counter measure studies to detect and deteriorate radars started too. This field is known as EW and is commonly defined as:

Electronic warfare is defined as any military action that protects the use of the electromagnetic (EM) spectrum, which includes the entire radio frequency (RF) spectrum, the infrared (IR) spectrum, visible spectrum, and ultraviolet (UV) spectrum, as well as direct energy for friendly purposes while denying it to the enemy [17]. Electronic warfare has three major subdivisions: Electronic Attack (EA), Electronic Protection (EP), and Electronic Support (ES) [1]. For more details on EW topic refer [1,18].

Radar Electronic Countermeasures (ECM) and ECCM were investigated in [19] where the historical development of ECM and ECCM was explained, and some major techniques were described. Depending on the jamming source, ECM techniques can be grouped as noise, repeater, and transponder jamming. Noise jamming is not a coherent jamming. With the development of ECCM features, these kinds of jamming techniques are no more effective for modern radars, which have coherent processing capability [18]. In the repeater jamming, victim radar signals are used for jamming after amplification and frequency modulation. Transponder Jamming is achieved by using a memory device that helps to record the incoming radar signal and regenerate it when needed. Initially, Frequency Memory Loop (FML) was used but it could not provide real coherent waveform, then DRFM was invented. DRFM can record the radar signal in a digital domain and modulate the signal, in time, frequency, and amplitude as desired and regenerate back when needed. Both repeater and transponder (with DRFM) jamming provide coherent jamming, so jamming will be very effective [20].

EW has been mainly considered for the military domain. If there would be a possibility of an intentional jamming against ADAS radars, considering the EW while developing an automotive radar would make them more robust. For an automotive radar design, EW is first to be considered in the EP domain to make them robust against intentional and unintentional jammers. Then, the detection of the jamming is the next step to counteract to the jamming.

As the ADAS radars apply coherent processing, jamming techniques also need to be coherent. Therefore, DRFM-based Coherent False Target (CFT) and Correlated Range and Velocity Gate Pull Off/In (RVGPO/I) techniques have been investigated on one of the ADAS radars, i.e., Long-Range Radar (LRR). The detail of CFT and RVGPO/I can be found in [1,2,3,4,5,6,7,8,9,10,11,12,13,14,15,16,17,18] and these techniques are considered as spoofing techniques.

### 2.3. DRFM

The DRFM is the indispensable part of the modern ECM systems as it provides the ability to generate coherent jamming waveforms which is a crucial means to jam advanced radars. DRFMs have been used in jammer systems for several decades now.

The DRFM samples incoming radar signals by using a high-speed Analog to Digital Converter (ADC), store-sampled data in the RF memory, applies the desired jamming modulation, and regenerates the digital signal into the analog signal by using a high-speed Digital to Analog Converter (DAC). The DRFM provides the ability to manipulate the incoming and stored signals in the amplitude, frequency, and phase to generate a wide variety of deception signals [20].

### 2.4. Jamming ADAS Radar Systems

To analyze the possibility of the jamming of the ADAS radars, some scenarios are defined against the LRR whereas the LRR has a longer range than any other radar in the ADAS which increases the jamming engagement time and therefore the possibility of being jammed. For short range radars, the jamming engagement time would be short. Jamming of the ADAS radars have been investigated in the following steps:Step-1: Define the assumptions and LRR parameters used on the analysis.Step-2: Calculate the necessary jamming power for the desired Jamming-to-Signal Ratio (JSR) on the radar detection.Step-3: Define jamming scenarios.Step-4: Investigate the jamming scenarios to find the necessary jamming power.Step-5: Investigate the effects of jamming on radar detection at radar Range Doppler Map (RDM) and 2-Dimensional Constant False Alarm Rate (2D-CFAR).Step-6: Analyze the possible jamming system.

#### 2.4.1. Step-1: Define the Assumptions and LRR Parameters Used on the Analysis

Important parameters of LRR used in the analysis are defined in Table 1 and Table 2.

Assumptions considered in the analysis are mentioned below:The LRR uses linear polarization as referred in Table 1. In case the orientation of the linear polarization is not known, the jammer must use circular polarization to improve the jamming chance with the loss of 3 dB.FMCW radars have high processing and modulation gains, so it is difficult to jam them with noncoherent jamming methods. The best jamming method is to use a DRFM to achieve a coherent jamming.For effective jamming, the jammer has to monitor radar parameters. In case there is a parameter update on the radar, then the jammer has to update the jamming parameters. Hence, the jammer needs a receiving cycle (look-through) and during that time the jamming could be interrupted which reduces the effects of the jamming. In this study, it is assumed that the jammer can receive and transmit simultaneously without any interruptions so that the jammer can follow radar parameters. For that purpose, the required isolation between jammer receiver and transmitter is calculated and presented below.The jamming power, detected by the victim radar antenna, is analyzed for four different scenarios initially. It is assumed that the jamming signals, which are generated by a DRFM and the jammer signals are up to date, are processed by the radar like real object echo signals. For one scenario, radar detection and processing with RDM and 2D-CFAR are investigated to inspect the expected outcome on the radar detection process.The DRFM memory size is not considered a problem for the jamming, especially for long pulses.The propagation delay of the jammer is not considered. There is a propagation delay in the jamming system. So, the jamming signal arrives later than the real echo signal. If the jammer can follow the radar parameter, with the appropriate timing, this delay can be compensated by the jammer.Because of the spatial diversity of a MIMO radar, it is difficult to jam them [23]. For RVGPO/I techniques, the jammer is assumed to be in the same angle with the manipulated real target.Compared to the radar signal, which matches with the radar receiver setting, the jammer signal has a certain amount of jamming losses. Jamming losses for a stationary jammer is 12 dB and for a mobile jammer, 9 dB are considered because of the following reasons:
(a)It is assumed that the stationary jammer stays on the roadside, and they are on the edge of the −3 dB beamwidth. For a mobile jammer case, no loss is considered as they might be on the boresight of the jammer antenna.(b)MIMO radars can generate simultaneously multiple transmit (Tx) beams [24]. Maximum coherency is achieved when all Tx channels transmit simultaneously using the same bandwidth [9]. However, we assumed that all channels transmit in different bandwidths to consider the worst case for the required jamming power. To jam all these signals simultaneously, jamming power split to four which results in 6 dB loss in the jammer output.(c)The polarization loss is taken as 3 dB.

The loss will be implemented in the related formula.

#### 2.4.2. Step-2: Calculate the Necessary Jamming Power for the Desired JSR on the Radar Detection

To calculate the necessary jamming Effective Radiated Power (ERP) for the related jamming scenarios, one-way and two-way Friis’ transmission equations are used [18] with the parameters defined in Table 1 and Table 2. One-way Friis’ transmission Equation (1) is used to find the received jamming signal power *Pr_J_*, and two-way Friis’ transmission equation is used find the received radar signal power from the object echo *Pr_R_*.
*Pr_J_* = (*P_J_ G_J_*)/(4*πR_J_*^2^) *A_e_* = (*ERP_J_*) (*λ*^2^
*G_R_*)/(4*πR_J_*)^2^(1)
where *A_e_* is the effective aperture area of the radar, *P_J_ G_J_* is the ERP of the jammer, *λ* is the wavelength of the radar signal, *G_R_* is the gain of the radar. *R_J_* is the distance of the jammer to the radar.
*Pr_R_* = (*P_R_ G_R_ σ*)/(4*πR_O_*^2^)^2^
*A_e_* = (*ERP_R_ σ λ*^2^
*G_R_*)/((4*π*)^3^(*R_O_*)^4^)(2)
where *P_R_ G_R_* is the ERP of the radar, *R_O_* is the distance of the object to the radar, and *σ* is the Radar Cross Section (RCS) of the object.
*JSR* = (*ERP_J_* 4*πR_O_*^4^)/(*ERP_R_ σ R_j_*^2^)(3)
*ERP_J_* = (*JSR ERP_R_ σ R_j_*^2^)/(4*πR_O_*^4^)(4)

Equations (1) and (2) are used to find the necessary *JSR* value as defined in Equation (3). To find the necessary *ERP_J_*, Equation (3) is reorganized in Equation (4). The loss factors mentioned above would be considered in the formula to obtain more realistic results.

Although the key factor in effective jamming technique is to keep the JSR as high as possible, the JSR ratio can be reduced to the 3 to 6 dB range for an effective ECM technique by using the coherent jamming waveforms as long as the jamming power is placed in the appropriate tracking gate of the victim radar [1,25]. During the dwell portion of the RVGPO/I techniques, a high JSR will increase the chance of stealing the gate. To be on the safe side, 10 dB JSR is considered for RVGPO/I techniques in our calculation.

For CFT techniques power calculation, the victim radars Minimum Discernable Signal (MDS) is an important parameter to consider. Whenever there is a signal over the MDS level, this could be considered as an object. MDS values can be predicted as typical noise floor plus Signal-to-Noise Ratio (SNR) of the radar. The SNR is dynamically set by the CFAR algorithm. However, for this study, JSR = 1 is taken to be on the safe side. This approach would require more jamming power.

#### 2.4.3. Step-3: Define Jamming Scenarios

The following five scenarios were investigated for the LRR jamming:

##### Scenario-1

In this scenario, a realistic false pedestrian echo is aimed to be generated with applying a CFT jamming technique by a jammer as shown in Figure 1 where there is no real object. However, with the successful jamming techniques, the radar will observe a false pedestrian or automobile echo.

##### Scenario-2

In this scenario, an RVGPO/I technique is applied by the jammer located in the automobile which is in front of the radar as shown in Figure 2. The aim is to manipulate the range and velocity information of the own platform detected by the LRR.

##### Scenario-3

In this scenario, an RVGPO/I technique is applied by the jammer as shown in Figure 3. In contrast to the Scenario-1, in Scenario-3, there is a real pedestrian, and the aim is to manipulate the parameters of the pedestrian which is measured by the radar.

##### Scenario-4

In this scenario, an RVGPO/I technique is applied by the jammer as shown in Figure 4. There is a real automobile, and the aim is to manipulate the parameters of the automobile which is measured by the radar.

#### 2.4.4. Step-4: Investigate the Jamming Scenarios to Find the Necessary Jamming Power

Required jammer ERP values with respect to the jamming range for each scenario are shown in Figure 5a,b, Figure 6 and Figure 7a,b, respectively. Figure 5b shows Scenario-1 with a ghost automobile echo generation. For all scenarios, except for Scenario-2, the required jammer ERP increases with the increasing jamming range. Contrary to the others, in Scenario-2, the jamming and the radar signal has the same range. For the radar signal power, the range has inverse 4th power factor and for the jammer signal power, the range has inverse 2nd power factor. Therefore, as the range increases, the required jamming power decreases for the desired JSR value.

Depending on the jamming scenario conditions, the required jamming ERP is changing from −2 dBm to 40 dBm. These ERP values are very low and easily realizable.

#### 2.4.5. Step-5: Investigate the Effects of Jamming on Radar Detection at Radar RDM and 2D-CFAR

Scenario-2 is chosen for further analysis to investigate the effects of jamming on the RDM and 2D-CFAR detection. Parameters of the scenario are defined in Table 3.

In the scenario, parameters of the LRR CFAR are defined as follows:Two-dimensional CFAR detectorProbability of False Alarm Rate (FAR) is %0.000000001 (1 × 10^−9^)Guard band size is 3 cellsTraining band size is 10 cells

The LRR radar signal waveform used in simulation is shown in Figure 8. The FMCW signal is chosen as a linear ramp for simplicity.

The automobile in front of the radar is detected by the LRR RDM and 2D-CFAR are shown in Figure 9a,b, respectively. The range and velocity (which have a relative speed of 10 km/h) are measured correctly.

When the jammer on the front automobile starts jamming, the LRR detects both jamming and echo signals. At first, the JSR is considered as 0 dB which means the jamming signal has the same power level of the echo signal. Figure 10a,b shows the RDM and 2D-CFAR detection, respectively, when the JSR is 1. The real echo is in the 50 m range, and the 10 km/h relative speed and ghost echo are in the 40 m range and 15 km/h relative speed in both RDM and 2D-CFAR figures.

When the jamming power level is increased to JSR = 10 dB, the LRR increases the detection threshold, and the real object could not be detected as the jamming signal has more power. Figure 11a,b shows the RDM and the 2D-CFAR detection, respectively, when the JSR is 10 dB. In this case, the real echo could not be detected by the CFAR. Only ghost echo is detected at a range of 40 m and relative speed of 15 km/h.

The RDM and the 2D-CFAR processing of the LRR shows the effect of jamming on the radar.

#### 2.4.6. Step-6: Analyze the Possible Jamming System

ERP: Considering these four scenarios, the required ERP is changing from −2 dBm to 40 dBm depending on the jamming scenario. With high gain antenna of 30 dBi and 10 mW transmitter, 40 dBm could be achieved. This shows that the required power level for the jammer is very low.

Sensitivity: The skin paint of the radar signal for different range values is shown in Figure 12a. The lowest value of the skin paint for a 500 m range is −54 dBm. Considering the worst-case conditions, a receiver with a 5 GHz instantaneous bandwidth (76–81 GHz) and 15 dB Noise Figure, the noise floor of a receiver is −62 dBm. To achieve −54 dBm sensitivity, there is still enough margin for SNR which is 8 dB. Polarization and atmospheric loss are not considered here.

Isolation: The value of isolation between the transmitter and the receiver provides an important information about the interoperability of Tx and Rx. If the isolation is bigger than the maximum Tx power minus the minimum Rx sensitivity, then there would be no interoperability problem. Therefore, Tx and Rx can operate at the same time and same frequency without interfering with each other. So, the ECM system can jam the radar with up-to-date radar signal information.

As shown in Figure 12b, the isolation requirement is less than 85 dB. To achieve 85 dB isolation, highly directive and carefully separated Tx and Rx antennas and isolation material can be used.

## 3. Results

In this paper we present the investigation of jamming scenarios and their possible effects on the ADAS radar applications. Our goal was to analyse the possibility of ADAS jamming and the required jammer features. In contrast with other works in the literature that investigate the ADAS jamming, we analyse the jamming system requirements and the effect of the jamming on the ADAS radar on the radar detection i.e., RDM and the 2D-CFAR detection. All possible jamming scenarios were investigated with coherent ECM techniques. While defining scenarios, LRR features and close to real-life conditions were taken into account. Scenario analyses are given in Table 4. Based on the described scenarios and analysis, it can be inferred that the ADAS radars can be jammed as also observed in [10,11,12,13]. The required jamming system features are not on the edge of the current technology and are easy to implement. To jam the ADAS radars, even −2 dBm ERP could be enough depending on the scenario. Moreover, the jamming power can easily be increased, which might result in a dramatic result on the ADAS radars performance.

We provided an analysis of the ADAS radar jamming phenomena with the EW perspective. It is known that the ADAS radars can be jammed. However, how and with what sort of jamming system the ADAS jamming can be achieved is not clear. We concluded that the ADAS jamming could be easily achieved. As the advancement on EW resulted in the implementation of ECCM features on the military radars, it is expected that the ADAS radars will also implement the ECCM features to make them less vulnerable to intentional jamming. On the other hand, implementation of complex ECCM features make the ADAS radars more expensive. Even the driving factor in the commercial market is costly, and the safety concerns always comes first then the cost. Then, it is expected that the ADAS radar manufacturers will soon consider the intentional jamming and implement radar ECCM features to minimize the vulnerability of the ADAS radars against intentional jamming.

## Figures and Tables

**Figure 1 sensors-22-06142-f001:**
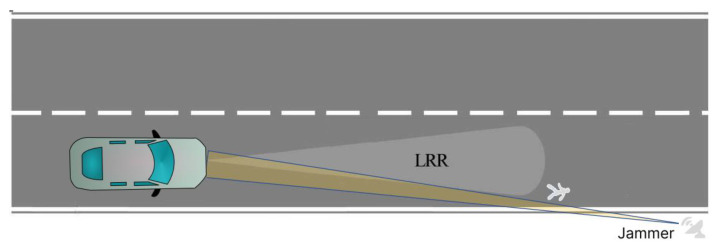
Scenario-1. A realistic false pedestrian echo generation by CFT.

**Figure 2 sensors-22-06142-f002:**
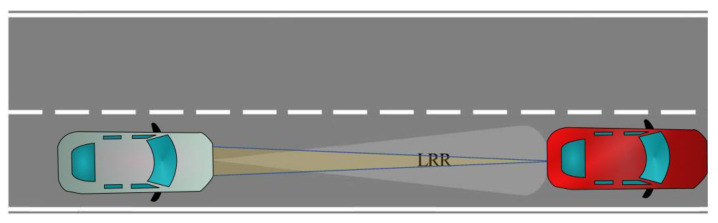
Scenario-2. An RVGPO/I technique for spoofing.

**Figure 3 sensors-22-06142-f003:**
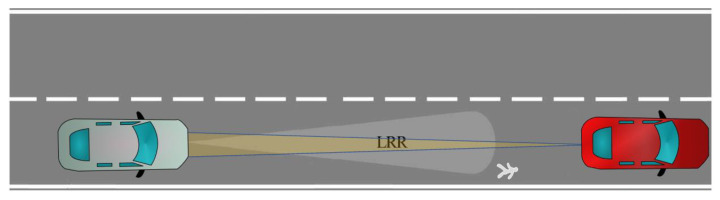
Scenario-3. An RVGPO/I technique for spoofing the real pedestrian echo.

**Figure 4 sensors-22-06142-f004:**
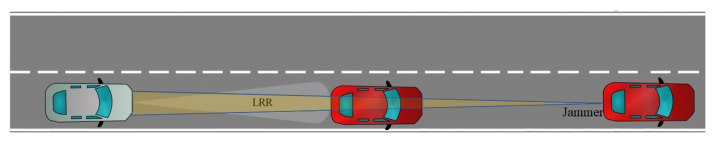
Scenario-4. An RVGPO/I technique for spoofing the real automobile echo.

**Figure 5 sensors-22-06142-f005:**
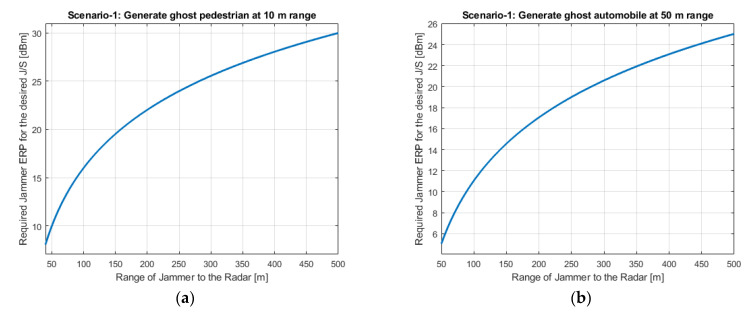
Jammer ERP versus range of the jammer for Scenario-1. (**a**) For the pedestrian false echo scenario; (**b**) For the automobile ghost echo scenario.

**Figure 6 sensors-22-06142-f006:**
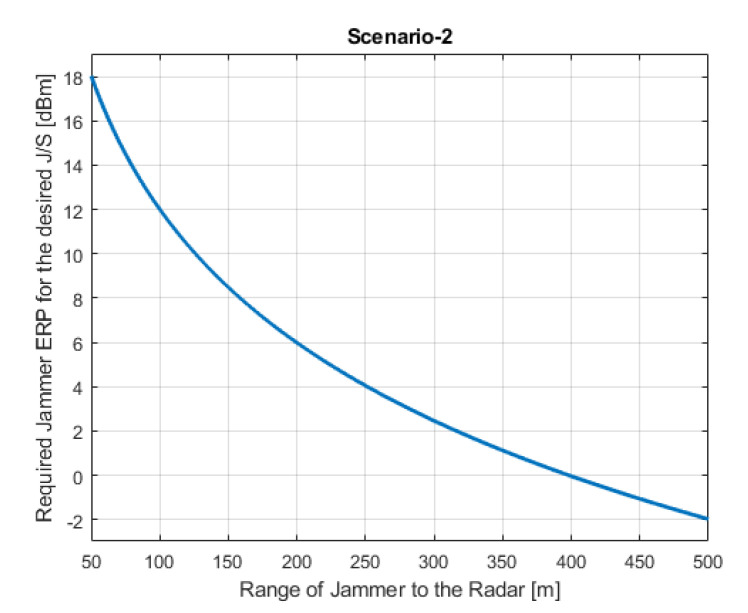
(Scenario-2) Jammer ERP versus range of the jammer where jammer is on the target object and applying RVGPO/I.

**Figure 7 sensors-22-06142-f007:**
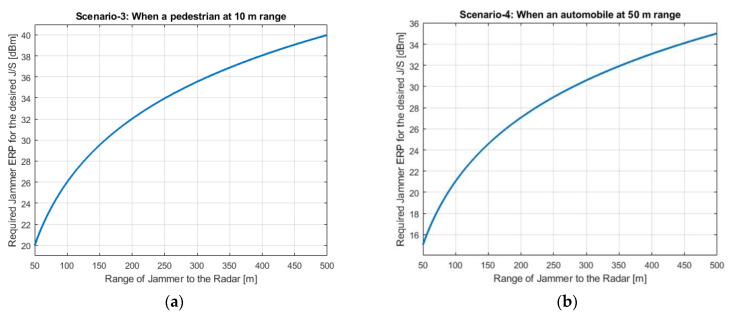
Jammer ERP versus range of the jammer. (**a**) Scenario-3: The pedestrian is at 10 m; (**b**) Scenario-4: The automobile is at 50 m.

**Figure 8 sensors-22-06142-f008:**
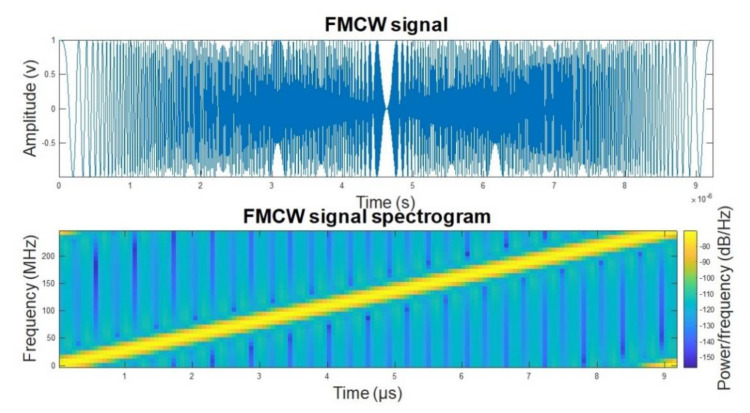
The LRR transmit signal waveform.

**Figure 9 sensors-22-06142-f009:**
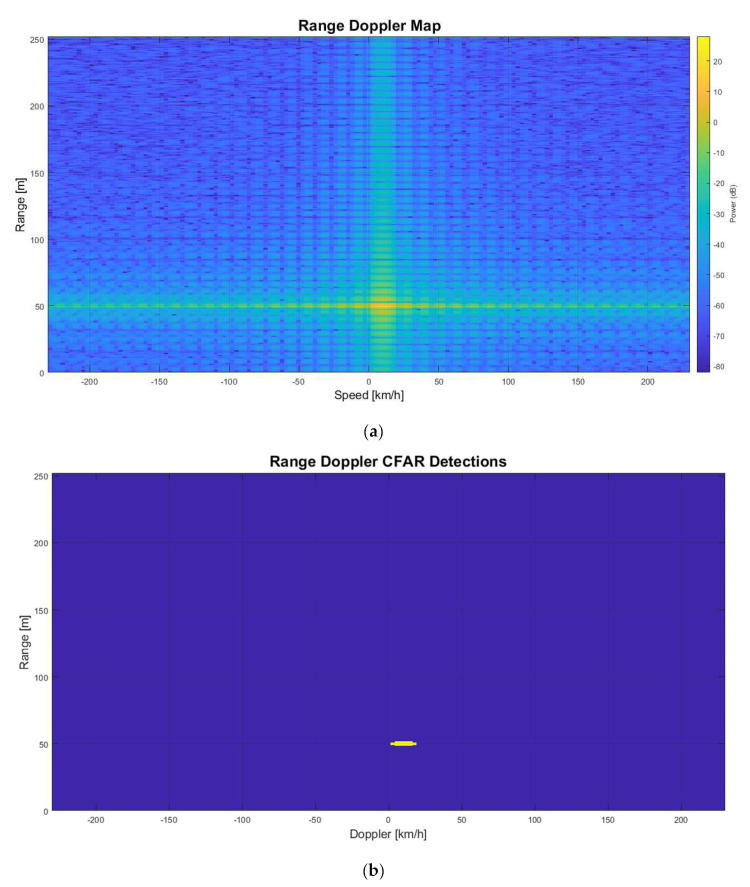
No jamming. (**a**) Range-Doppler Map; (**b**) 2D-CFAR Detection.

**Figure 10 sensors-22-06142-f010:**
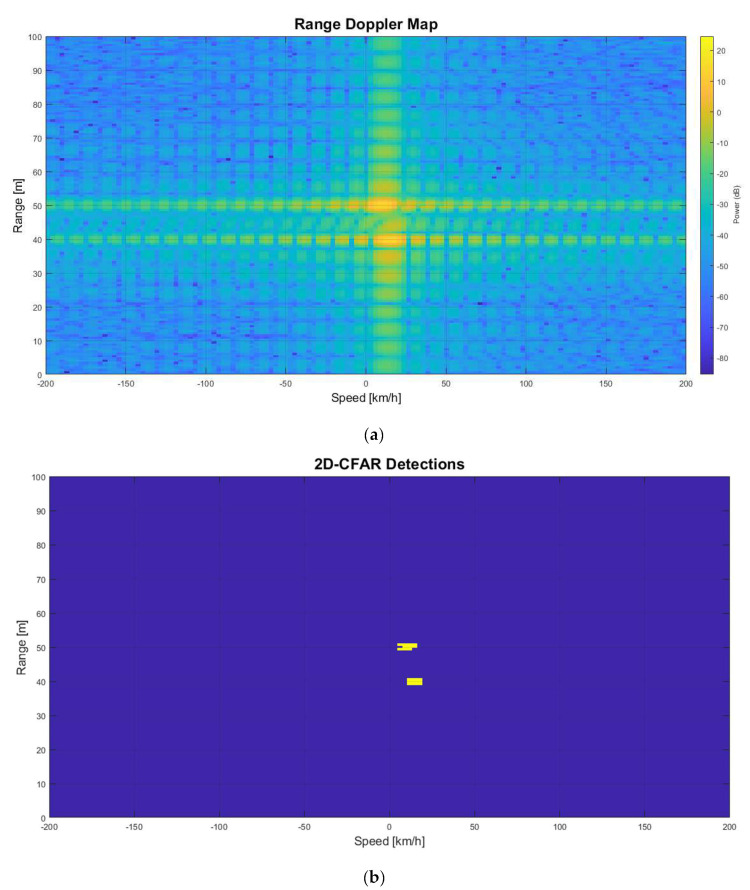
Jamming with JSR = 1. (**a**) Range-Doppler Map; (**b**) 2D-CFAR Detection.

**Figure 11 sensors-22-06142-f011:**
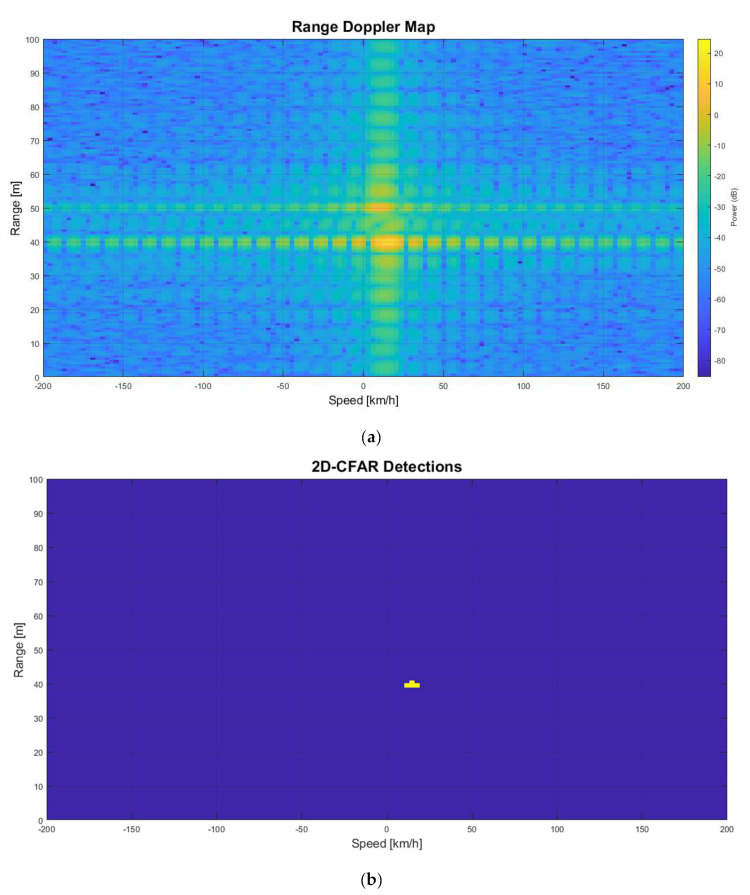
Jamming with JSR = 10. (**a**) Range-Doppler Map; (**b**) 2D-CFAR Detection.

**Figure 12 sensors-22-06142-f012:**
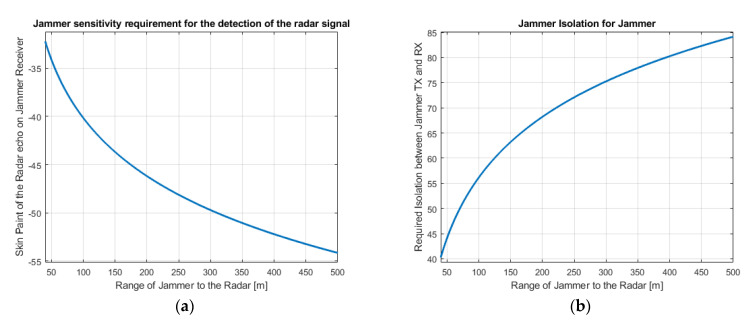
(**a**) The skin paint of the radar signal; (**b**) Required isolation level between the Rx and Tx of the jammer.

**Table 1 sensors-22-06142-t001:** LRR parameters [21].

LRR Parameters	Values
LRR max range	250 m
Tx/Rx antenna max gain (single element)	20–25 dBi
Tx/Rx antenna beamwidth (−3 dB) Azimuth	± 5°
Tx power ERP (peak)	30–40 dBm (40 dBm is used here)
Rx typical Noise Figure	15 dB
Rx IF typical noise floor	−99 dBm/MHz
Rx IF bandwidth (After signal processing)	500 kHz–25 MHz
Antenna polarization	Horizontal, Vertical, Diagonal
Interference, clutter removal technique	CFAR

**Table 2 sensors-22-06142-t002:** Radar Cross Section values [22].

Object	RCS Values
Automobile	19–25 dBm^2^ (20 dBm^2^ used here)
Pedestrian	−3 dBm^2^

**Table 3 sensors-22-06142-t003:** Scenario parameters for RDM and 2D-CFAR detection.

Parameters	Values
Relative distance of the real object	50 m
Relative distance of the false echo generated by jamming	45 m
The speed of radar platform	100 km/h
The speed of real object	90 km/h
The speed of false echo	85 km/h
Desired JSR	10 dB

**Table 4 sensors-22-06142-t004:** Scenario analysis.

Scenario	Expected Result	Effect Level
1. Generate false pedestrian/automobile echo by CFT	False echo is detected by the LRR, and the ADAS is expected to act. Depending on the false echo parameters, emergency brake might be applied.	Driving quality reduced. Depending on the circumstances, an accident might happen.
2. Deception jamming with RVGPO/I	The real object information could not be detected. Depending on the jamming direction: RVGPO: The LRR measures farther distance than the real value. RVGPI: The LRR measures shorter distance than the real value.	Driving quality reduced. RVGPO: The LRR perceives longer free space in front of the automobile, in case the detection threshold increased, and the real object cannot be detected anymore. In this scenario jammer platform can also be affected so it would be considered as not a realistic scenario. RVGPI: The LRR perceives shorter free space in front of the automobile, and act to decrease the speed of the platform. Depending on the circumstances, an accident may happen.
3–4. Deception Jamming. Manipulate the measured radar parameters for the pedestrian/automobile which is already detected by the radar.	Jamming signal with higher JSR, increases the detection threshold and hide the real echo. If the techniques are effective, then the radar will detect only the jammer signal. Depending on the jamming signal, the LRR provides wrong information about the pedestrian/automobile. RVGPO: The LRR measures the pedestrian/automobile farther and did not react in time. RVGPI: The LRR measures shorter distance than the pedestrian/automobile and apply emergency brake.	RVGPO: The LRR perceives longer free space in front of the automobile, in case the detection threshold increased, and the real object cannot be detected anymore. In this scenario jammer platform can also be affected so it would be considered as not a realistic scenario.

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
