# Peer review of "Analysis of ADAS Radars with Electronic Warfare Perspective"

_sensors, 2022, doi:10.3390/s22166142_

Round 1

Reviewer 1 Report

This research is very interesting, only includes real-world applications in a Smart City as in:

Ingi GudmundssonGregory Falco:
Porting Computer Vision Models to the Edge for Smart City Applications: Enabling Autonomous Vision-Based Power Line Inspection at the Smart Grid Edge for Unmanned Aerial Vehicles (UAVs). HICSS 20221-10

and with another applications to a Smart City as in:

https://www.amazon.com.mx/Innovative-Applications-Smart-Cities-Alberto/dp/036782096X

Reviewer 2 Report

My comments are as follows:

 1.        In the abstract part, the novelty and key idea of the proposed method should be described. The authors only described that “In this paper, we present the investigation of Advanced Driver Assistance Systems (ADAS) radars in the perspective electronic warfare (EW)”. The novelty and key idea are not clear.

2.        In abstract, the new finding of this work should be described briefly data. The authors only described that “The proposed results present the examination and comparison of different scenarios based on simulations and measurements or ADAS performance”. I can understand what the authors did. But I cannot understand the new finding of this work.

3.        The authors should not quote Ref. [1] in the abstract part.

4.        The quotation of previous articles is rough. For example, “Moreover, there is a growing interest in intentional jamming of ADAS radars in the literature [11-17]”. (See p. 2.) These citations are meaningless. The authors must quote articles properly.

5.        Due to so general research survey, the problem definition of this work is not clear. In the introduction part, the drawbacks of each conventional technique should be described one by one. The authors should emphasize the difference with other methods to clarify the position of this work further.

6.        In the Introduction part, the new features of the proposed approach and the main advantages of the results over others should be clearly described. The reviewer fails to understand the novelty of the proposed approach. As you know, ADAS test is not novel.

7.        In the Introduction part, strong points of this proposed method should be further stated and organization of this whole paper is supposed to be provided in the end.

8.        Do you have a permission to use all figures? For example, Fig. 1. If not, this infringes the copyright of other journals.

9.        In section 2.4.2, some equations are presented. However, this paper is lack of rigorously theoretic derivation. The author needs to clarify the theoretical derivation or cite some related articles using these equations.

10.     Please unify the font style. In sentences/equations, mathematical expressions should be Italic font. (Some of them are Italic fonts and others are Roman font.) Otherwise, readers will be confused. For example, see Eq. (1).

11.     The authors should use clear images for Fig. 9.

12.     The new finding of this work is not clear. The authors should clearly describe the new finding and scientific contribution of this work by comparing with the state-of-the-art articles. Several articles are discussed in the research survey. However, no comparison is shown with these articles. Frankly speaking, the research survey and references are meaningless.

13.     There is no conclusion part of this paper. The authors should summarize the result of this paper briefly in the conclusion part.

Reviewer 3 Report

1. In line 9, the style of some fonts is different from others. 2. The abstract should be re-written to introduce more information about the study conducted by the author, not just the background. 3. The abbreviation should be defied when they are firstly used, e.g. EW in line 52. 4. In section 2.4.1, the author just shows the model parameters and some assumptions, but there’s no details about the LRR model, which is necessary to evaluate this study. Otherwise, the title of section 2.4.1 should be improved. 5. The style of variables is different in equations and main text. 6. As mentioned in lines 208, PrJ and PrR are calculated by (1) and (2) using the parameters defined in Table 1 and 2, but the values of most parameters in (1) and (2) cannot be found in Table 1 and 2, e.g., Gr, ERPJ and etc. Moreover, many parameters defined in Table 1 and 2 are not used in (1) and (2). 7. In section 2.4.2, it is suggested to give the calculation equation directly with the reference, because this model is not presented by the author. 8. In line 229, the author mentioned that JSR=1 is taken, then why using (1)-(3)? The correctness of the calculation model for the jamming power is doubtful. 9. In section 2.4.3, why choosing the four scenarios? Is there any special consideration? 10. The contribution of this study is not clear. It seems that the author just used the existing models to calculate the jamming power under some scenarios. I cannot find any new method, technique or founding.

Round 2

Reviewer 2 Report

In this paper, the authors presented the investigation of Advanced Driver Assistance Systems (ADAS) radars in the perspective electronic warfare (EW). Overall, the authors have made a good attempt, I think. In the revised version, most of the reviewer’s requests were met by the authors. The reviewer would like to pay tribute to the authors’ great work. This is scientifically sound and contains sufficient interest to merit publication, I think.

Reviewer 3 Report

No more comments.